# Absolute and Relative Reliability of Spatiotemporal Gait Characteristics Extracted from an Inertial Measurement Unit among Senior Adults Using a Passive Hip Exoskeleton: A Test–Retest Study

**DOI:** 10.3390/s24165213

**Published:** 2024-08-12

**Authors:** Cristina-Ioana Pîrșcoveanu, Anderson Souza Oliveira, Jesper Franch, Pascal Madeleine

**Affiliations:** 1Department of Health Science and Technology, ExerciseTech, Aalborg University, 9260 Gistrup, Denmark; jfranch@hst.aau.dk (J.F.); pm@hst.aau.dk (P.M.); 2Department of Materials and Production, Aalborg University, 9220 Aalborg, Denmark; oliveira@mp.aau.dk

**Keywords:** accelerometer, cadence, speed, step length, walking

## Abstract

Background: Seniors wearing a passive hip exoskeleton (Exo) show increased walking speed and step length but reduced cadence. We assessed the test–retest reliability of seniors’ gait characteristics with Exo. Methods: Twenty seniors walked with and without Exo (noExo) on a 10 m indoor track over two sessions separated by one week. Speed, step length, cadence and step time variability were extracted from one inertial measurement unit (IMU) placed over the L5 vertebra. Relative and absolute reliability were assessed using the intraclass correlation coefficient (ICC), standard error of measurement (SEM) and minimal detectable change (MDC). Results: The relative reliability of speed, step length, cadence and step time variability ranged from “almost perfect to substantial” for Exo and noExo with ICC values between 0.75 and 0.87 and 0.60 and 0.92, respectively. The SEM and MDC values for speed, step length cadence and step time variability during Exo and noExo were <0.002 and <0.006 m/s, <0.002 and <0.005 m, <0.30 and <0.83 steps/min and <0.38 s and <1.06 s, respectively. Conclusions: The high test–retest reliability of speed, step length and cadence estimated from IMU suggest a robust extraction of spatiotemporal gait characteristics during exoskeleton use. These findings indicate that IMUs can be used to assess the effects of wearing an exoskeleton on seniors, thus offering the possibility of conducting longitudinal studies.

## 1. Introduction

Reliable assessment of gait characteristics in real-world scenarios using inertial measurement units (IMUs) is crucial to ensure consistent and accurate gait data [1,2,3,4,5]. Accurate spatiotemporal gait characteristics of senior adults are crucial in clinical applications to distinguish between measurement errors or relevant biological variations related to age and pathologies [6]. Although “almost perfect” reliability of spatiotemporal gait characteristics like walking speed, cadence and step length extracted from a single IMU placed on the lower back is reported in the literature concerning senior adults with pathologies [1,2,3,7,8], understanding the reliability of gait characteristics of healthy senior adults [1,6,9,10,11] provides the basis for comprehending age-related biomechanical changes [6] and the relevance of intervention methods to preserve and promote physical activity [12]. Despite the large number of studies in the field, reliability studies investigating how newer devices such as exoskeletons affect the gait of senior adults are still lacking.

Common spatiotemporal gait characteristics associated with age are diminished walking speed, reduced step length [13] and elevated cadence [14,15]. Wearable devices like exoskeletons have been used to assist and, therefore, influence spatiotemporal gait characteristics [16]. Exoskeletons can be medical or non-medical devices [17] with multiple classifications based on the power supply (active and passive) and targeted segment (upper and lower body) [18,19]. Wearing a lower-limb passive exoskeleton has been shown to modify the gait characteristics of senior adults by increasing their walking speed and step lengths, as well as reducing their cadence when walking at their preferred walking speed [20]. Limited research has been dedicated to investigating the reliability of wearing exoskeletons, and the subsequent impact of such technology on gait characteristics, validly and reliably [21,22,23]. However, most of the research has focused on the reliability of the mechanical device to deliver assistance [21,23] with no focus on the reliability of the device for human usage or the repeatability of human–machine interaction. Therefore, it is relevant to examine the spatiotemporal gait parameters extracted from IMUs of senior adults to ascertain the reliability of the data when utilizing a passive exoskeleton.

The Mobilise-D consortium identified and developed algorithms for estimating gait detection events and validated such algorithms on several senior adult subpopulations [3]. Walking speed, cadence and stride length were the three gait characteristics that were validated by the algorithm [3]. Even when using a pre-validated model on similar populations, gait characteristics’ individuality [24], sensors’ susceptibility to drift [25] and false step detection (detecting non-gait activity like arm movement as gait events) [26] can occur. Therefore, further research is warranted to test the reliability of the gait detection extraction procedure in various setups and conditions.

The main objective of the present study was to assess the relative and absolute test–retest reliability of senior adults’ spatiotemporal gait characteristics extracted from IMU recordings of senior adults using a passive hip exoskeleton using the Mobilise-D pre-validated model. This research contributes to the existing body of knowledge by conducting an investigation into the reliability of gait parameters among senior adults when utilizing or not utilizing a passive exoskeleton.

## 2. Materials and Methods

This test–retest reliability study followed the guidelines for reporting reliability and agreement studies (GRRAS) [27]. In agreement with GRRAS, the sample selection, study design and statistical analysis are reported in the Appendix A.

### 2.1. Participants

Twenty senior adults (seventeen females and three males aged 72.3 ± 4.5 years; height 1.7 ± 0.1 m and body mass 76.2 ± 13.9 kg) participated in the study. The participants included were able to walk unassisted and had no uncorrected visual or hearing impairments. Participants were excluded if they presented with a history of neurological disorders or chronic back and lower-extremity pain [28]. All participants were recruited from local senior activity centers and provided written informed consent. The study followed the guidelines of the ethics committee of the North Denmark Region (LBK nr. 1083) and was in agreement with the Helsinki Declaration.

### 2.2. Study Protocol

In this test–retest reliability study, participants were invited to attend two indoor laboratory sessions held seven days apart. Each participant had to walk as naturally as possible in the following two conditions: with a bilateral passive hip exoskeleton (Exo) and without the device (noExo). The order of the conditions was randomized and balanced across all participants, but the same order was used for the retest. Similarly, the same exoskeleton was used during the test–retest sessions. A 10 min warm-up walk and a 5 min familiarization period of walking with the Exo were provided for each participant [20]. All participants were provided with a standard black T-shirt and shorts. The participants wore their own shoes (the same shoes for both sessions).

The gait characteristics of the participants were recorded on an indoor obround walkway consisting of two 10 m straight walkways and two 2 m turns (Figure 1A). The participants were instructed to walk at their preferred walking speed between two pre-established points. They were asked to wait in the anatomical standing position, which was maintained for approximately two seconds. At this time, they would be given an auditory cue which signaled the start and end of each trial. If the participants completed the walkway before the cue, they were instructed to wait until the second audio cue. After the second cue, they returned to the start point, which concluded the recording of one trial. The inter-cue duration was determined during the familiarization period, where the average walking time used to complete the walkway was monitored and noted (10–15 s). The quiet stance sequences and keeping the anatomical position between the audio cues were later used in post-processing to detect when the participant walked forward from the start to the stop point and walked back to the start point (Figure 1B), thus marking each trial. Five walking trials were recorded for each condition.

### 2.3. Gait Analysis

The participants were fitted with an IMU (AX3, Axivity, Newcastle, UK) that recorded triaxial accelerations at 100 Hz. The IMU was fixed to the lower back at the level of the fifth lumbar vertebrae (L5) location using tape [2]. The IMU data were recorded offline using the OpenMovement GUI application (OmGUI v1.0.0.43, Newcastle University, Newcastle, UK) and stored in the RAM of the device. Before the start of each walking condition, the participants were asked to jump or squat three times (if jumping was too challenging). The events were used to determine the change in the vertical acceleration and to detect the start of each condition (Exo and noExo).

### 2.4. Exoskeleton

The aLQ (IMASEN Electrical Industrial Co. Ltd., Aichi, Japan), a bilateral passive exoskeleton, was used in the present study. The Exo has an adjustable waist belt and bilateral spring mechanisms which clip onto the belt and are strapped 12 cm above the knee joint using Velcro straps. The Exo is categorized as “unpowered” and uses the pendulum motion model to aid walking by loading the spring mechanism during hip extension and releasing it during hip flexion [20,29] (Figure 1A). The Exo has a range of motion of 84° and a torque of ≤2.8 Nm. The total weight of the system is approximately 600 g.

### 2.5. Data Analysis

The IMU data were downloaded via the OpenMovement GUI application in csv format and imported into MATLAB^®^ (MathWorks, Release R2023b v23., Natick, MA, USA) as a variable containing timestamps, raw mediolateral, horizontal–vertical and anterior–posterior accelerations. The acceleration data were filtered using a 4th-order Butterworth filter with a cutoff frequency of 20 Hz in agreement with previous studies [2,3,30]. The order of the conditions and the height of the participant were imported separately using an Excel file (Microsoft 365 Apps v 2110, Redmond, WA, USA) to segment and reorder the data into conditions. For spatiotemporal parameter estimations, the data were converted from g to m/s^2^.

The vertical acceleration data were used to estimate the following spatiotemporal gait characteristics: cadence, step length, walking speed and step time variability [31]. A pre-validated gait detection algorithm from the Mobilise-D consortium available online on GitHub [3] was adapted to fit our dataset and used for the calculations. The specific code modifications made can be found in the Appendix A. Furthermore, the step time variability was extracted based on Whitlle’s calculations [32]. Only the steps collected between the audio cues, while walking forward, were included in the statistical analysis (Figure 2B).

### 2.6. Statistics

An average range of 63 to 77 steps was analyzed for each participant and each condition (Exo: test—64 ± 22 steps; retest—67 ± 27 steps; noExo: test—69 ± 30 steps; retest—77 ± 40 steps). The relative reliability was assessed with the intra-class correlation coefficient (ICC_2,1_ for absolute agreement) and has been computed for each outcome. The ICC_2,1_ was interpreted as “poor” for values ≤ 0.20, as “fair” for values ≥ 0.21 and ≤0.40, as “moderate” for values ≥ 0.41 and ≤0.60, as “substantial” for values ≥ 0.61 and ≤0.80, and as “almost perfect” for values ≥ 0.81 and ≤1.00 [33]. The absolute reliability [4] was estimated for each outcome by calculating the standard error of measurement (*SEM*) using the following equation:(1)$$SEM=SD\sqrt{1-ICC}$$
where *SD* represents the standard deviation of each gait characteristic in each condition in both sessions. Furthermore, the minimal detectable change (*MDC*) representing the minimal value to recognize a real difference was calculated using the following equation [34]:(2)$$MDC=SEM*1.96*\sqrt{2}$$

A paired Student’s *t*-test between sessions (test–retest) for each condition was conducted in IBM SPSS Statistics (v29, IBM SPSS^®^ Statistics, Armonk, NY, USA), where the significance level was set at *p* < 0.05. Before the statistical test, the normality of data was visually inspected using Q-Q plots and box plots and assessed by the Shapiro–Wilk test [35]. Differences were expressed as means and confidence intervals (CI) of 95%. Furthermore, Bland–Altman plots were used to assess the heteroscedasticity of the measurements and to detect possible bias in mean differences between test and retest (test value−retest value) for each gait characteristic parameter and condition. Bias is present when the retest measurement results in a distinct mean, while heteroscedasticity manifests when the mean difference between measurements varies in relation to the average [36].

## 3. Results

All gait spatiotemporal characteristics were normally distributed except walking speed in the noExo condition (*p* < 0.03). However, when examining the Q-Q plots, the data indicated normal distribution, and no further action was taken.

The relative and absolute reliability of gait characteristics are reported in Table 1. The ICCs for cadence, step length and walking speed were categorized as “almost perfect”, with the ICC values ranging from 0.86 to 0.92 for the noExo condition. Slightly smaller ICCs, though still within the “almost perfect” category, were seen for the Exo condition with values between 0.81 and 0.87 for the same parameters. The step time variability showed “substantial” ICC values for Exo (0.60) and noExo (0.75). The SEM values and MDC values for cadence were between 0.12 and 0.30 steps/min and 0.32 and 0.83 steps/min, respectively. A similar trend was seen for step time variability, where the SEM values and MDC values were between 0.1 s and 0.39 s and 0.27 s and 1.06 s, respectively. Smaller values (≤0.01 m and ≤0.01 m/s) for both SEM and MDC were seen for step length and speed. The gait characteristics (cadence, walking speed and step length) did not present any statistical differences between sessions for either condition (Table 1). However, significant differences were found for step time variability between the two sessions, both during Exo and noExo.

The analysis of the Bland–Altman plot for cadence (Figure 2A) indicated a systematic difference between the test and the retest for the noExo condition with a bias of −1.6 steps/min (lower cadence values at test compared with retest). During the Exo condition, the analysis of the Bland–Altman plot for cadence (Figure 2B) also indicated a systematic difference between the test and the retest sessions with a bias of 0.5 steps/min (higher cadence values at test compared with retest). Most of the individual differences between test and retest fall within the expected range of agreement. The plots show two outliers (one in noExo and one in Exo), suggesting an inconsistency of one participant in each condition. It is noteworthy that the outliers did not influence the overall bias and were two different participants. Similarly, the analysis of the Bland–Altman plots for step time variability (Figure 2G,H) shows systematic differences between test and retests for both noExo and Exo conditions with a bias of 1.21 s and 4.74 s, respectively. Most of the individual differences between test and retests of the step time variability fall within the expected range of agreement for the noExo condition, with only two outliers. However, the plot for the Exo condition indicates eight outliers, suggesting much more inconsistency with the individual data. The analysis of the Bland–Altman plots for step length (Figure 2C,D) and speed (Figure 2E,F) did not indicate any systematic bias between the test and retest sessions in either the noExo or Exo conditions.

## 4. Discussion

This study provides novel insight into the reliability of IMU-derived spatiotemporal gait characteristics of healthy senior adults wearing a passive hip exoskeleton. The cadence, step length and gait speed of senior adults showed “almost perfect” test–retest reliability during both Exo and noExo, though slightly less reliability was noted for Exo. “Substantial” test–retest reliability was seen for the step time variability in both walking conditions, with slightly higher ICC values for Exo. Additionally, the SEM was below 0.002 m for step length, below 0.002 s speed and below 0.3 step/min for cadence in both walking conditions, indicating precise measurements between the test and retest sessions. Similar patterns were seen for the MDC. The findings of this study suggest that both Exo and noExo yield consistent results in terms of spatiotemporal gait characteristics for senior adults derived from the Mobilise-D consortium. However, other gait characteristics that were not included in the consortium like the step time variability need to be further investigated. These findings demonstrate that common gait characteristics of healthy senior adults are almost identical when walking with and without a passive hip exoskeleton using a single IMU. Hence, this Mobilise-D method provides a solid foundation for evaluating gait characteristics when wearing a hip exoskeleton in the elderly population. The measurements employed may be further utilized to accurately assess the effectiveness of interventions using an exoskeleton for an extended period of time to improve spatiotemporal gait characteristics among senior adults.

Limited information is available on the reliability of spatiotemporal gait characteristics among healthy seniors. The ICC values we found for cadence, step length and gait speed (0.81–0.92) were slightly smaller than the values reported by Hartmann et al. [1] which ranged from 0.92 to 0.96 during overground walking on a gym floor extracted from an IMU positioned correspondingly at L5. A similar ICC value of 0.94 was reported by Faude et al. [10] for stride length using photoelectric timing gates. Compared with the present study, low ICC values of 0.28 and 0.65 were reported for step length by Almarwani et al., 2016 [6] and Galna et al., 2013 [37] in healthy senior adults and Parkinson’s patients, respectively, when gait characteristics were measured using pressure mats. Lower ICC values of 0.51 and 0.63 for step time variability were reported by Del Din et al., 2016 [2] for healthy senior adults and Parkinson’s patients, respectively, using GaitRite and lower-back IMU. The higher values of step time variability found in the present study may be explained by the difference in the protocols, as Del Din et al. [2] used straight-line walking, compared with continuous walking, which may have allowed for better gait variability estimation [37]. Furthermore, the review by Kobsar et al., 2020 [7] on the use of IMU’s ability to reliably measure gait characteristics revealed good to poor validity for extracting step length and step time variability as well as excellent reliability of step length estimations. However, the study also featured a lack of consistency in reporting results, which poses challenges in comparing findings across studies. Feodoroff and Blümer [20] observed significant improvements in step length and a reduction in cadence in their study, contrary to the trends seen in our study, with a reduction in step length and speed and increased cadence. However, there are significant distinctions between their study and the one being discussed here. Firstly, Feodoroff and Blümer 2022 [20] focused on gait-impaired patients, while the current study revolves around asymptomatic senior adults. These contrasting populations exhibit distinct gait patterns, making direct comparisons impossible. Secondly, Feodoroff and Blümer 2022 [20] applied the exoskeleton unilaterally on the affected joint and compared it with the unaffected joint. Conversely, the present study involves a bilateral application of the exoskeleton. The unilateral use of the exoskeleton may result in asymmetric gait patterns, leading to disparate outcomes. Although the two studies used the same exoskeleton, due to the difference in population and application of the exoskeleton, a direct comparison cannot be made.

While this study focused on the reliability of some spatiotemporal gait characteristics of healthy senior adults during Exo and noExo walking conditions, it is crucial to acknowledge that the familiarity and individual comfort of wearing the exoskeleton may influence the gait characteristics. Therefore, the generalization of these findings to a broader population (i.e., patients or other age ranges) is not possible, as the exoskeleton may affect the wearer in different ways. Feodoroff and Blümer 2022 [20] investigated the efficacy of a passive exoskeleton among individuals with compromised walking abilities. Their results revealed notable enhancements in step length along with lowered cadence, signifying the advantageous nature of such devices for individuals with impaired walking patterns [20]. However, Pirscoveanu et al., 2022 [29] did not see any changes in spatiotemporal parameters while using the exoskeleton in able-bodied adults, suggesting that the aid provided by the exoskeleton differs between able-bodied and impaired adults. To the authors’ knowledge, no other study has investigated the reliability of spatiotemporal gait characteristics in senior adults (with or without pathologies) when walking with a passive exoskeleton. However, the use of exoskeletons, both powered and passive, as walking aid devices for seniors is not only gaining popularity among rehabilitation centers but also among the senior population in general [38]. Such devices may also be used to motivate and stimulate an active lifestyle within the senior population and help this segment of the population meet the WHO recommendations for physical activity. As exoskeleton development focuses on seniors’ needs, it is imperative to understand not only the immediate effect that such technology can provide, but also the long-term effects.

A strength of this study was the use of a low-cost technology (IMU) to extract spatiotemporal gait characteristics for test–retest assessments of walking with and without a passive hip exoskeleton among senior adults. Our investigation not only emphasized the reliable performances of the technology, but also shed light on the accelerometer’s potential applicability on a broader scale as a digital health tool that may be incorporated into exoskeletons. As non-obstructive wearable devices, accelerometers allow for continuous monitoring of gait parameters in real-life settings. These devices are currently being implemented on a larger scale as health-monitoring tools for senior adults [3,9,39] and have the potential to be integrated into the design of walking aids and exoskeletons. This development will allow for continuous monitoring of the physical activity of the wearer as well as detection of possible falls or decreases in mobility. Nonetheless, the extent to which a wearable device can accurately identify changes in gait characteristics, potentially indicating a deterioration or pathology in the locomotor system or merely reflecting the natural aging process, requires further longitudinal investigation.

While accelerometer data derived from IMUs can provide reliable gait analysis for senior adults, relying solely on this technique presents limitations that must be acknowledged. The exclusion of inclinometer and gyroscope measurements restricts the findings of the present study by failing to include information regarding postural changes and rotational movements. Furthermore, it is important to acknowledge the limitations of the model implemented, as it is based on the inverted pendulum model and has the following two assumptions: compass gait cycle and straight-line walking at a constant pace, which can affect the estimation of the vertical displacement (Del Din et al., 2016) [2]. Lastly, it is important to acknowledge that gait analysis encompasses various other gait parameters beyond the scope of this study. Parameters like gait variability and asymmetry, double support time and foot cycle duration can indicate the presence of gait impairments [40] as well as age-related declines [15]. However, the model used in the present study has only been validated for cadence, step length and speed. Nevertheless, it is noteworthy that the inclusion of an additional parameter, namely step time variability (a measure of gait stability), yielded inconclusive results and unreliable data. Therefore, future studies should take into consideration the advantages of utilizing multiple sensors and incorporating additional gait characteristics to ensure an overall assessment of human locomotion.

## 5. Conclusions

The present study demonstrated high-to-moderate test–retest relative and absolute reliability of senior adults walking with and without a passive hip exoskeleton spatiotemporal gait characteristic, collected by an IMU. The relative reliability of cadence, speed and step length exceeded the threshold of 0.81, whereas the step time variability showed values between 0.60 and 0.75, indicating a strong to moderate agreement between repeated measurements. The small SEM and MDC values of cadence, speed and step length further support the ability of the approach to detect real gait changes. Moreover, the observed statistical discrepancies in step time variability between the test–retest sessions for Exo and noExo indicate the necessity of incorporating an alternative approach for evaluating this parameter. Overall, these findings contribute to the understanding of the reliability of step length, cadence, speed and step time variability in senior adults when wearing a passive hip exoskeleton. Furthermore, this study provides the opportunity to explore the potential longitudinal repercussions of utilizing passive hip exoskeletons in senior populations.

## Figures and Tables

**Figure 1 sensors-24-05213-f001:**
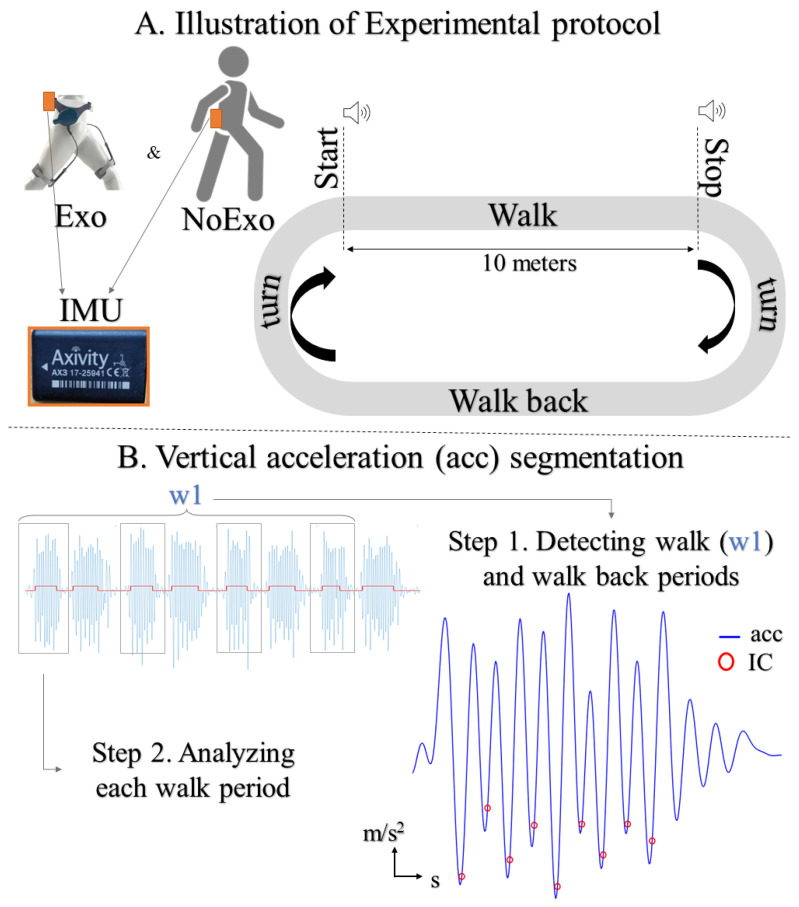
(**A**)—Illustration of experimental design and (**B**)—vertical acceleration segmentation procedure used to collect and analyze the gait of senior adults, where acc is vertical acceleration, IC is foot initial contact, and W1 is walk forward.

**Figure 2 sensors-24-05213-f002:**
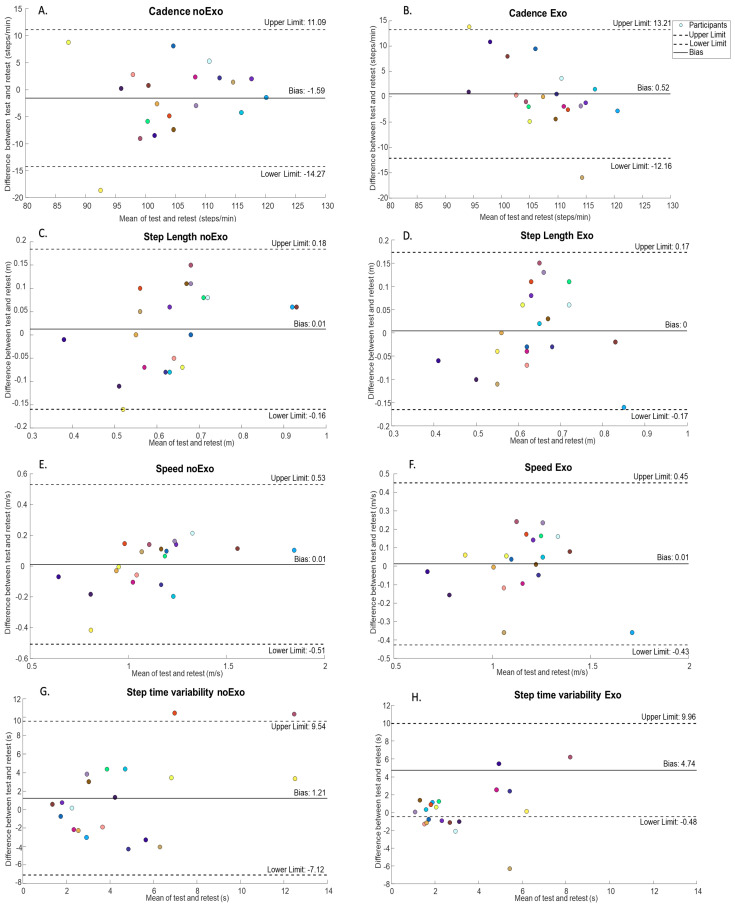
Bland-Altman plots for (**A**,**B**) cadence, (**C**,**D**) step length and (**E**,**F**) speed and step time variability (**G**,**H**) where the left and right panels show the data for participants walking without wearing the exoskeleton (noExo) condition and walking while wearing the exoskeleton (Exo), respectively. The upper and lower limits are illustrated by dashed lines and the bias as a solid full line. Each participant is illustrated as an unique color within the graphs.

**Table 1 sensors-24-05213-t001:** Test–retest reliability of spatiotemporal gait characteristics (cadence, step length, walking speed and step time variability) in senior adults when wearing or not wearing a passive hip exoskeleton (Exo/noExo) while walking at preferred speed (N = 20). CI: confidence interval; Exo: exoskeleton; ICC: intra-class correlation coefficient; MDC: minimal detectable change; noExo: no exoskeleton; SEM: standard error of measurement.

Outcomes	Condition	Test Mean (95% CI)	Retest Mean (95% CI)	ICC_2,1_ (95% CI)	SEM ^1^	MDC ^2^	*t*-Test
Cadence (steps/min)	noExo	104.13 (99.57; 108.69)	105.72 (101.66; 109.79)	0.86 (0.65; 0.94)	0.298	0.827	*t*(19) = −1.1, *p* = 0.29
	Exo	107.79 (105.00; 110.57)	107.27 (102.84; 111.69)	0.81 (0.50; 0.92)	0.115	0.320	*t*(19) = 0.36, *p* = 0.72
Step length (m)	noExo	0.65 (0.58; 0.72)	0.64 (0.58; 0.69)	0.88 (0.71; 0.95)	0.002	0.005	*t*(19) = 0.59, *p* = 0.56
	Exo	0.64 (0.59; 0.69)	0.63 (0.59; 0.68)	0.82 (0.55; 0.93)	0.001	0.002	*t*(19) = 0.20, *p* = 0.85
Walking speed (m/s)	noExo	1.13 (0.99; 1.28)	1.12 (1.01; 1.23)	0.92 (0.79; 0.97)	0.001	0.004	*t*(19) = 0.27, *p* = 0.79
	Exo	1.15 (1.04; 1.26)	1.14 (1.03; 1.25)	0.87 (0.66; 0.95)	0.002	0.006	*t*(19) = 0.31, *p* = 0.76
Step time	noExo	5.25 (3.28; 7.31)	3.33 (1.82; 4.25)	0.60 (0.06; 0.84)	0.383	1.062	*t*(19) = 2.20, *p* = 0.04
variability (s)	Exo	4.04 (2.90; 5.36)	2.95 (2.05; 3.75)	0.75 (0.35; 0.90)	0.097	0.268	*t*(19) = 2.41, *p* = 0.03

^1^ using pooled data and ^2^ at the 95% CI interval.

## Data Availability

The datasets presented in this article are not readily available because of confidentiality agreements. Requests to access the datasets should be directed to the corresponding author.

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
