# Peer review of "Absolute and Relative Reliability of Spatiotemporal Gait Characteristics Extracted from an Inertial Measurement Unit among Senior Adults Using a Passive Hip Exoskeleton: A Test–Retest Study"

_sensors, 2024, doi:10.3390/s24165213_

Round 1
Reviewer 1 Report
Comments and Suggestions for Authors
This manuscript is about the measurement reliability of elderly walking using IMU sensors attached to an exoskeleton. IMU sensors (accelerometers) are already implemented in various devices in the healthcare field, and the main discussion theme of this manuscript is "originality and novelty."
Below are my review comments:
#1 The introduction is insufficient, and I cannot understand the novelty and originality of this study. Moreover, as mentioned in the following comments, the authors state that "it is only reported in the literature on ~," but in reality, research in this field is advancing.
#2 Line 44 In addition to the waist-worn IMU, there are many papers that have examined the walking characteristics of the elderly. https://www.mdpi.com/1424-8220/20/10/2858 https://link.springer.com/article/10.1186/s12984-020-00685-3
#3 Figure 1-B Red circles representing Initial Contact are indicated on the signal waveform, but they are not identifiable even when significantly enlarged.
#4 Line 113 Readers may find it difficult to obtain information about aLQ, a Japanese product, so I recommend that the authors include more information about the exoskeleton. The current information provided does not give readers sufficient details about the exoskeleton. For example, consider including that aLQ is "unpowered," its "weight," and that it "assists the pendulum motion model during walking."
Author Response
The authors would like to thank the reviewer for the insightful comments and suggestions that have helped us improve the overall quality of the manuscript. Please find the detailed responses below and the corresponding revisions highlighted in blue in the re-submitted files.
Comment #1 The introduction is insufficient, and I cannot understand the novelty and originality of this study. Moreover, as mentioned in the following comments, the authors state that "it is only reported in the literature on ~," but in reality, research in this field is advancing.
Response #1: The novelty of our study relies on establishing reliability when performing gait analysis using a single IMU in older adults using a passive exoskeleton. Despite the large number of studies in the field, reliability studies are lacking, and our study is a major contribution to the field. It is difficult to interpret the comment "it is only reported in the literature on ~," as the manuscript does not comprise this specific sentence. Published peer-reviewed papers are cited in the section, but the focus of the introduction (see also responses to the queries of the other reviewers) section has been changed to reflect the application and impact of passive hip exoskeletons on senior’s mobility highlighting the relevance of determining the reliability of gait parameters extracted from IMU sensor data in such conditions. (changes can be found P1-2, L33-73 marked in blue)
Comment #2 Line 44 In addition to the waist-worn IMU, there are many papers that have examined the walking characteristics of the elderly. https://www.mdpi.com/1424-8220/20/10/2858 https://link.springer.com/article/10.1186/s12984-020-00685-3
Response #2: The suggested papers have been added to the manuscript. (changes can be found P1, L39-40, marked in blue)
Comment #3 Figure 1-B Red circles representing Initial Contact are indicated on the signal waveform, but they are not identifiable even when significantly enlarged.
Response #3: The figure has been amended, and Figure 1-B panel has been moved below Figure 1-A panel and the IC figure has been enlarged. (changes can be found P3, between L96 and 97).
Comment #4 Line 113 Readers may find it difficult to obtain information about aLQ, a Japanese product, so I recommend that the authors include more information about the exoskeleton. The current information provided does not give readers sufficient details about the exoskeleton. For example, consider including that aLQ is "unpowered," its "weight," and that it "assists the pendulum motion model during walking."
Response #4: We included some further information regarding the system while still respecting our confidentiality agreement with the company. Due to confidentiality agreements, the information that we are allowed to disclose on the device is limited, however, we did specify the weight of the device, mentioned it is unpowered, and that it assists by using the pendulum motion model. We hope to meet the reviewer requirement with these changes. (changes can be found P4, L124-130 marked in blue)

Reviewer 2 Report
Comments and Suggestions for Authors
The article by PîrÈ™coveanu et al. is well-written and their description of the methodology is remarkable. It demonstrates good levels of test-retest reliability for an IMU (AX3 Axivity) in a senior adult population under two conditions (wearing or not wearing an exoskeleton).
In my opinion, the study's objective is not clear. The participants do not have any pathology and have not been recommended to use an exoskeleton for ambulation, so the practical applications and impact on current knowledge of the conclusions are limited. Additionally, the reliability and validity of this sensor have already been studied in populations with specific pathologies and previously compared to other systems. I believe the authors should clarify their hypothesis and how it will contribute to existing knowledge. If the study aims to focus on the validation of the sensor, this point should be made clear in all sections, from the title to the conclusions. If the focus is on changes due to the exoskeleton, the statistical analysis should be reconsidered. Finally, if the goal is to evaluate the reliability of the sensor while wearing an exoskeleton, a population that has been prescribed this device should be selected, not healthy senior adults.
Furthermore, I agree with the authors that reducing gait analysis to cadence, step length, and speed is overly simplistic. In fact, knowing step length and cadence allows one to directly calculate speed, so the study actually analyzes the reliability of a sensor that has already been previously studied, under the condition of wearing an exoskeleton in people who have not been recommended to use one, and only for two variables. I believe the authors should delve deeper into their analyses and clarify this if they aim to have their article continue through the review process.
The discussion section is not very reflective and does not make a real comparison with previous studies on the validity and reliability of this sensor in other populations. Not only should other studies with photoelectric timing gates and pressure mats be commented on, but several existing articles (including some review articles) on the concurrent validity of different commercially available IMUs should be included. Additionally, the possible differences in the obtained results should be discussed, providing justifications or possible explanations for these discrepancies. I believe this chapter should be expanded and thoroughly worked on by the authors, focusing on the validity of this device.
Minor comments:
The abstract needs to be better written. what does "both Exo" refer to?
Line 139: The sentence has no sense. Check it, please.
Line 161: Please remove the instructions.
Author Response
The authors would like to thank the reviewer for the insightful comments and suggestions that have helped us improve the overall quality of the manuscript. Please find the detailed responses below and the corresponding revisions highlighted in blue in the re-submitted files.
Comment #1: In my opinion, the study's objective is not clear. The participants do not have any pathology and have not been recommended to use an exoskeleton for ambulation, so the practical applications and impact on current knowledge of the conclusions are limited. Additionally, the reliability and validity of this sensor have already been studied in populations with specific pathologies and previously compared to other systems. I believe the authors should clarify their hypothesis and how it will contribute to existing knowledge. If the study aims to focus on the validation of the sensor, this point should be made clear in all sections, from the title to the conclusions. If the focus is on changes due to the exoskeleton, the statistical analysis should be reconsidered. Finally, if the goal is to evaluate the reliability of the sensor while wearing an exoskeleton, a population that has been prescribed this device should be selected, not healthy senior adults.
Response #1: The use of exoskeletons in the senior population is not only met with skepticism by healthcare professionals, but it is extremely hard to be accepted by seniors. The purpose of the device tested in our study is to help seniors preserve their physical activity and serve as an alternative walking aid. As such, testing on healthy seniors, is highly relevant, as the application of such devices on more injury-prone populations may be unethical and require extensive ethical guidelines. Furthermore, most passive exoskeletons have not been classified as medical devices, some can be purchased ‘off-the-shelf’ and may or may not have a targeted population.
The objective of the present study has been improved to highlight the reliability of passive exoskeletons used by senior adults extracted using a pre-validated algorithm. Applying the same algorithm to a different population and in various conditions will impact the output and as such the reliability of the algorithm needs to be reconfirmed under the new conditions. These aspects have been improved in the introduction. (changes can be found P1-2, L33-73, marked in blue)
Comment #2: Furthermore, I agree with the authors that reducing gait analysis to cadence, step length, and speed is overly simplistic. In fact, knowing step length and cadence allows one to directly calculate speed, so the study actually analyzes the reliability of a sensor that has already been previously studied, under the condition of wearing an exoskeleton in people who have not been recommended to use one, and only for two variables. I believe the authors should delve deeper into their analyses and clarify this if they aim to have their article continue through the review process.
Response #2: Similar analysis and statistical approaches were found in the literature (Attias et al 2016, Graser et al 2022) but only comparing the exoskeleton conditions. The no exoskeleton condition in our manuscript brings necessary information on the test-retest gait characteristics of the participants in a standard control situation which is valuable information for future randomized control trials in elderly populations. The IMU is a small unit, highly suitable to attach to the human body in combination with larger objects such as an exoskeleton. Unfortunately, the limitations of a triaxial accelerometer make much deeper analysis such as step width and step width variability challenging. However, stride time variability, an important measure of gait stability (Svenningsen et al 2020), has been added to the manuscript (Whittle, 2007). (changes can be found P5-7, L173-213 marked in blue)
Comment #3: The discussion section is not very reflective and does not make a real comparison with previous studies on the validity and reliability of this sensor in other populations. Not only should other studies with photoelectric timing gates and pressure mats be commented on, but several existing articles (including some review articles) on the concurrent validity of different commercially available IMUs should be included. Additionally, the possible differences in the obtained results should be discussed, providing justifications or possible explanations for these discrepancies. I believe this chapter should be expanded and thoroughly worked on by the authors, focusing on the validity of this device.
Response# 3: The discussion section has been improved to provide comparisons with other studies focusing on the reliability of gait characteristics and wearables. The differences between our obtained results have been highlighted and justified as well as the limitations of the present study have been added. (changes can be found P8-10 , L215-330)
Minor comments
Comment #4: The abstract needs to be better written. what does "both Exo" refer to?
Response #4: The abstract has been improved and a clear distinction between the two walking conditions is now present. (changes can be found P1, L15-29, marked in blue)
Comment #5: Line 139: The sentence has no sense. Check it, please.
Response #5: The sentence’s readability has been improved. (changes can be found P4, L150-154, marked in blue)
Comment #6: Line 161: Please remove the instructions.
Response #6: The instructions have been removed.
Additional references used in the replies to the comments
Attias, M., Bonnefoy-Mazure, A., De Coulon, G., Cheze, L., & Armand, S. (2016). Feasibility and reliability of using an exoskeleton to emulate muscle contractures during walking. Gait & Posture, 50, 239-245.
Graser, J. V., Prospero, L., Liesch, M., Keller, U., & van Hedel, H. J. (2022). Test–retest reliability of upper limb robotic exoskeleton assessments in children and youths with brain lesions. Scientific Reports, 12(1), 16685.
Svenningsen, F. P., Pavailler, S., Giandolini, M., Horvais, N., & Madeleine, P. (2020). A narrative review of potential measures of dynamic stability to be used during outdoor locomotion on different surfaces. Sports Biomechanics, 19(1), 120-140.
Whittle, M. W. (2014). Gait analysis: An introduction. Butterworth-Heinemann.

Reviewer 3 Report
Comments and Suggestions for Authors
Please review lines 58-59 as in ref 20, preferred walking speed and fast speed were different testing conditions.
In the introduction section, the authors mentioned that "Wearing a lower-limb passive exoskeleton has been shown to modify the gait characteristics of senior adults by increasing walking speed and step lengths, as well as reducing cadence". In table 1 the the mean cadence decreases, the mean step length decreases and the gait speed increases from noExo to Exo conditions. Will the authors discuss this?
While the paper aims to assess the reliability of gait spatiotemporal parameters when walking wearing a lower-limb passive exoskeleton, a big part of the discussion is written about the effect of wearing the Exo on the gait parameters. Will the authors reevaluate these parts?
Author Response
The authors would like to thank the reviewer for the insightful comments and suggestions that have helped us improve the overall quality of the manuscript. Please find the detailed responses below and the corresponding revisions highlighted in blue in the re-submitted files.
Comment #1: Please review lines 58-59 as in ref 20, preferred walking speed and fast speed were different testing conditions.
Response #1: Indeed ref 20 (Feodoroff and Blümer 2022) have tested two walking conditions, preferred and fast walking speed and the authors saw an increase in step length and a decrease in cadence in both walking speeds. The lines were revised to reflect this distinction, however, still focusing on the preferred walking speed (changes can be found P2, L50-53, marked in blue)
Comment #2: In the introduction section, the authors mentioned that "Wearing a lower-limb passive exoskeleton has been shown to modify the gait characteristics of senior adults by increasing walking speed and step lengths, as well as reducing cadence". In table 1 the the mean cadence decreases, the mean step length decreases and the gait speed increases from noExo to Exo conditions. Will the authors discuss this?
Response #2: The reviewer is right, the present findings indicate a different trend than the one described by Feodoroff and Blümer (2022). Although Feodoroff and Blümer (2022) saw improvements in step length and decreases in cadence, there are two major differences between the before-mentioned study and the present study: 1) the population used by Feodoroff and Blümer (2022) are orthopedic and neurologic patients with gait impairments, whereas the population used in the present study is able-bodied senior adults. The two populations have different gait patterns and cannot be directly compared and 2) Feodoroff and Blümer (2022) applied the device unilaterally and compared between the affected and unaffected legs, whereas the device is applied bilaterally in the present study. The application of the device unilaterally may create an unbalanced and asymmetric gait which will yield different outcomes. Thus, this discrepancy has been addressed in the discussion. (changes can be found P8-9, L 251-263, marked in blue)
Comment #3: While the paper aims to assess the reliability of gait spatiotemporal parameters when walking wearing a lower-limb passive exoskeleton, a big part of the discussion is written about the effect of wearing the Exo on the gait parameters. Will the authors reevaluate these parts?
Response #3: The discussion has been rewritten to demonstrate the reliability of spatiotemporal parameters and how they are affected by wearing an exoskeleton. (changes can be found P8-10 , L215-316 marked in blue)
Additional references used in the replies to the comments
Feodoroff, B., & Blümer, V. (2022). Unilateral non-electric assistive walking device helps neurological and orthopedic patients to improve gait patterns. Gait & Posture, 92, 294-301.

Round 2
Reviewer 1 Report
Comments and Suggestions for Authors
I have reviewed the manuscript as revised by the author.
It has been appropriately revised in response to my peer review comments.
I wish the author further development of his research.
Author Response
The authors would like to thank the reviewer for the insightful comments and suggestions that have helped us improve the overall quality of the manuscript.
Reviewer 2 Report
Comments and Suggestions for Authors
The authors have made a commendable effort to clarify all the points highlighted by this reviewer. I am now satisfied with the changes made.
Author Response

(The authors gave the same response as above.)
